# VoxAct-B: Voxel-Based Acting and Stabilizing Policy for Bimanual Manipulation

**I-Chun Arthur Liu**    **Sicheng He**    **Daniel Seita**$^{*}$    **Gaurav S. Sukhatme**$^{*\dagger}$
Department of Computer Science, University of Southern California

**Abstract:** Bimanual manipulation is critical to many robotics applications. In contrast to single-arm manipulation, bimanual manipulation tasks are challenging due to higher-dimensional action spaces. Prior works leverage large amounts of data and primitive actions to address this problem, but may suffer from sample inefficiency and limited generalization across various tasks. To this end, we propose VoxAct-B, a language-conditioned, voxel-based method that leverages Vision Language Models (VLMs) to prioritize key regions within the scene and reconstruct a voxel grid. We provide this voxel grid to our bimanual manipulation policy to learn acting and stabilizing actions. This approach enables more efficient policy learning from voxels and is generalizable to different tasks. In simulation, we show that VoxAct-B outperforms strong baselines on fine-grained bimanual manipulation tasks. Furthermore, we demonstrate VoxAct-B on real-world `Open Drawer` and `Open Jar` tasks using two UR5s. Code, data, and videos are available at https://voxact-b.github.io.

## 1 Introduction

Bimanual manipulation is essential for robotics tasks, such as when objects are too large to be controlled by one gripper or when one arm stabilizes an object of interest to make it simpler for the other arm to manipulate [1]. In this work, we focus on asymmetric bimanual manipulation. Here, "asymmetry" refers to the functions of the two arms, where one is a *stabilizing* arm, while the other is the *acting* arm. Asymmetric tasks are common in household and industrial settings, such as cutting food, opening bottles, and packaging boxes. They typically require two-hand coordination and high-precision, fine-grained manipulation, which are challenging for current robotic manipulation systems. To tackle bimanual manipulation, some methods [2, 3] train policies on large datasets, and some exploit primitive actions [4, 5, 6, 7, 8, 9, 10]. However, they are generally sample inefficient, and using primitives can hinder generalization to different tasks as they are not easily adaptable to other types of tasks.

To this end, we propose VoxAct-B, a novel voxel-based, language-conditioned method for bimanual manipulation. Voxel representations, when coupled with discretized action spaces, can increase sample efficiency and generalization by introducing spatial equivariance into a learned system, where transformations of the input lead to corresponding transformations of the output [11]. However, processing voxels is computationally demanding [12, 13]. To address this, we propose utilizing VLMs to focus on the most pertinent regions within the scene by cropping out less relevant regions. This substantially reduces the overall physical dimensions of the areas used to construct a voxel grid, enabling an increase in voxel resolution without incurring computational costs. To our knowledge, this is the first study to apply voxel representations in bimanual manipulation. Additionally, VoxAct-B does not rely on action primitives, making it more general and applicable to a wider range of tasks.

We also employ language instructions and VLMs to determine the roles of each arm: whether they are *acting* or *stabilizing*. For instance, in a drawer-opening task, the orientation of the drawer and

---

$^{*}$Equal advising
$^{\dagger}$GSS holds concurrent appointments as a Professor at USC and as an Amazon Scholar. This paper describes work performed at USC and is not associated with Amazon.

8th Conference on Robot Learning (CoRL 2024), Munich, Germany.

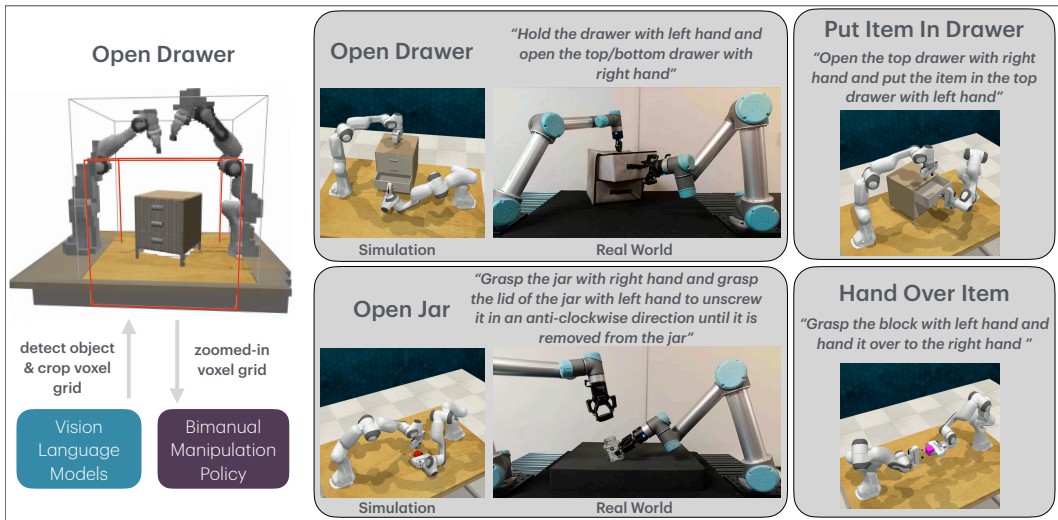

Figure 1: VoxAct-B uses voxel representations and language to perform bimanual manipulation with 6-DoF manipulation from both arms. We test four language-conditioned bimanual tasks in simulation and two (`Open Drawer` and `Open Jar`) on a real-world setup with two UR5s.

the position of the handle affect which arm is more suitable for opening the drawer (acting) and which is better for holding it steady (stabilizing). We use VLMs to compute the pose of the object of interest relative to the front camera and to decide the roles of each arm. Then, we provide appropriate language instructions to the bimanual manipulation policy to control the acting and stabilizing arms.

We extend the RLBench [14] benchmark to support bimanual manipulation. We introduce a bimanual version of `Open Drawer`, `Open Jar`, `Put Item in Drawer`, and `Hand Over Item` tasks. VoxAct-B outperforms strong baselines, such as ACT [3], Diffusion Policy [15], VoxPoser [16], and PerAct [11], by a large margin. We also validate our approach on a real-world bimanual manipulation setup with two UR5s on `Open Drawer` and `Open Jar`. See Figure 1 for an overview.

The contributions of this paper include:

- VoxAct-B, a novel method for bimanual manipulation which uses VLMs to reduce the size of a voxel grid for learning with a modified, downstream voxel-based behavior cloning method [11].
- A suite of vision-language bimanual manipulation tasks, extended from RLBench [14].
- Simulation experiments indicating that VoxAct-B achieves state-of-the-art results on these tasks.
- Demonstrations of VoxAct-B on a real-world bimanual manipulation setup with two UR5s.

## 2   Related Work

**Bimanual Manipulation.** There has been much prior work in bimanual manipulation for folding cloth [5, 8, 17, 18, 19, 20, 21], cable untangling [6], scooping [9], bagging [22, 23, 24], throwing [25], catching [26], and untwisting lids [27]. Other works study bimanual manipulation with dexterous manipulators [28, 29, 30, 31] or mobile robots [32]. In contrast to these works, our focus is on a *general approach* to bimanual manipulation with parallel-jaw grippers on fixed-base manipulators. Works that study general approaches for bimanual manipulation include [2, 4, 7, 33], which use primitive actions or skills to reduce the search space across actions. Other general approaches focus on orthogonal tools such as interaction primitives [34] or screw motions [35]. Recently, Zhou et al. [3] introduced another general approach, based on "action chunking" to learn high-frequency controls with closed-loop feedback and applied their method on multiple asymmetric bimanual manipulation tasks using low-cost hardware. Other works extended this by either incorporating novel imitation learning algorithms [36] or enhancing the hardware itself [37, 38]. However, these works may still require substantial training data and lack spatial equivariance for generalization. In closely-related work, Grannen et al. [39] decouple a system into stabilizing and acting arms

to enable sample-efficient bimanual manipulation with simplified data collection. While effective, this formulation predicts top-down keypoints and was not tested with 6-DoF manipulation. In contrast, our method supports 6-DoF manipulation for bimanual manipulation tasks. In independent and concurrent work, Grotz et al. [40] propose PerAct$^2$, extending RLBench to bimanual manipulation with 13 new tasks and presenting a language-conditioned, 6-DoF, behavior-cloning agent. This work is complementary to ours, and future work could merge techniques from both works.

**Action Space Representation.** For 2D manipulation, prior works have shown the benefits of action representations based on spatial action maps [21, 41, 42, 43, 44, 45, 46], including in bimanual contexts [10, 21], where neural networks directly predict 2D "images" that indicate desired locations for the action. Compared to directly regressing the action location, using spatial action maps better handles multimodality and has 2D equivariance, where translations and rotations of the input image map to similar transformations of the output action. Recent works have extended this idea to support 3D spatial action maps, which classify an action's location as a 3D point in the robot's workspace, and thus maintain spatial equivariance. For example, PerAct [11] is a language-conditioned behavioral cloning agent that takes voxel grids as input and outputs 6-DoF actions. While PerAct achieved state-of-the-art performance on RLBench, it has a high computational cost due to processing voxels. Follow-up works, such as RVT [12] and Act3D [13], have reduced the computational cost of PerAct by avoiding voxel representations but often need multiple views of the scene to achieve optimal performance and may be less interpretable compared to a voxel grid that contains a 3D spatial action map. These prior works have not been applied to bimanual manipulation. In this work, we retain the spatial equivariance benefits of voxel representations but reduce the cost of processing voxels by "zooming" into part of the voxel grid. This is similar to the intent of C2F-ARM [47] and RVT-2 [48], but we use the knowledge in VLMs to determine the most relevant regions in the voxel grid.

**LLMs and VLMs for Robotics.** LLMs and VLMs, such as GPT-4 [49], Llama 2 [50], and Gemini [51], have revolutionized natural language processing, computer vision, and robotics due to their strong reasoning and semantic understanding capabilities. Consequently, recent work has integrated them in robotics and embodied AI agents, typically as a high-level planner [52, 53, 54, 55], which may also produce code for a robot to execute [56, 57, 58]. We defer the reader to [59, 60, 61] for representative surveys. Among the most relevant prior works, Huang et al. [16] propose VoxPoser, which uses pre-trained LLMs and VLMs to compose 3D affordance maps and 3D constraint maps, which are then used with motion planning to generate trajectories for robotic manipulation. By leveraging LLMs and VLMs, VoxPoser can generalize to open-set instructions and objects. However, as we later demonstrate in experiments, VoxPoser can struggle with tasks that require high precision and contact. In this work, we demonstrate how to use VLMs to effectively process the input of PerAct for bimanual manipulation, obtaining the generalization benefits of VLMs with the precision capabilities of PerAct. In recent and near-concurrent work, Varley et al. [62] also uses VLMs for bimanual manipulation. Our work differs in that we do not fix the roles of each arm; we use an off-the-shelf VLM [63] without any fine-tuning, and we do not use a skills library.

# 3 Problem Statement

Given access to a pre-trained VLM and expert demonstrations, the objective is to produce a bimanual policy $\pi$ for a variety of language-conditioned manipulation tasks. We assume a flat workspace with two fixed-base robot manipulators, each with a parallel-jaw gripper. A policy $\pi$ controls both arms by producing actions $a_t = (a_t^s, a_t^a)$ at each time step $t$, where $a_t^s$ and $a_t^a$ follow [39] and refer to the *stabilizing* and *acting* arm actions, respectively. For simplicity, we suppress the time $t$ when the distinction is unnecessary. We use the low-level action representation from PerAct [11] with $a^s = (a_{\text{pose}}^s, a_{\text{open}}^s, a_{\text{collide}}^s)$ and $a^a = (a_{\text{pose}}^a, a_{\text{open}}^a, a_{\text{collide}}^a)$. These specify each arm's 6-DOF gripper pose, its gripper open state, and whether a motion planner for the arms used collision avoidance to reach an intermediate pose. We assume task-specific demonstrations $\mathcal{D}_\ell = \{\zeta_1, \zeta_2, \ldots, \zeta_n\}$ and two common language commands $\ell_{as}$ and $\ell_{sa}$, where $as$ denotes the left arm as acting and right arm as stabilizing, and vice versa for $sa$. Each demonstration consists of a set of keyframes extracted from a sequence of continuous actions paired with observations. We adapt the keyframe extraction function

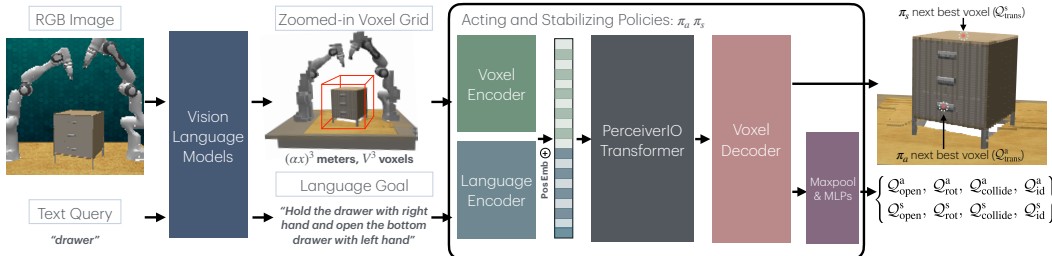

Figure 2: Overview of VoxAct-B. Given RGB-D images and a language goal, we input an RGB image from the front camera and a text query extracted from the language goal into the Vision Language Models (VLMs). The VLMs output the pose of the object of interest with respect to the front camera. This information determines the language goal and the roles of each arm (i.e., *acting* or *stabilizing*). Additionally, we use the object's position with the RGB-D images to reconstruct a voxel grid that spans $\alpha x^3$ meters of the workspace using $V^3$ voxels. The zoomed-in voxel grid, the language goal, proprioception data of both robot arms, and an arm ID are provided to an acting policy $\pi_a$ and a stabilizing policy $\pi_s$. The policies predict the discretized pose of the next best voxel, gripper open action, collision avoidance flag, and arm ID for fine-grained bimanual manipulation.

from [11] by including keyframes that have an action with near-zero joint velocities and unchanged gripper open state for acting and stabilizing arms. The observation at each time is the 3D voxel grid $\mathbf{v}$ of dimension $(L \times W \times H)$, where we use $\mathbf{v}[x, y, z]$ to denote an individual voxel at coordinates $(x, y, z)$. The voxel grid is reconstructed from RGB-D sensors. The robot also receives the language command $\mathbf{l} \in \{\ell_{as}, \ell_{sa}\}$, which is fixed for all time steps in an *episode*, where the robot interacts with the environment for up to $T$ time steps. An episode terminates with a task-dependent success criteria or failure (if otherwise).

## 4 Method

### 4.1 Extending PerAct for Bimanual Manipulation

PerAct [11] was originally designed and tested for single-arm manipulation. We extend it to support bimanual manipulation. A natural way to do this would be to train separate policies for the two arms. However, we exploit the discretized action space that predicts the next best voxel with spatial equivariance properties and formulate a system that uses acting and stabilizing policies. In contrast to a policy that operates in joint-space control, acting and stabilizing policies perform the same functions irrespective of whether it is a left arm or a right arm, assuming the next best voxel is kinematically feasible for both arms. This policy formulation enables more efficient learning from multi-modal demonstrations compared to a joint-space control policy. In the low-level action space, the arms execute one low-level action $a_t^s$ and $a_t^a$ (see Section 3) at each time $t$. In the following, we use similar notation as [11] but index components as belonging to an arm using the superscript: arm $\in \{$acting, stabilizing$\}$.

At each time step, the input to each arm is a voxel observation $\mathbf{v}$, proprioception data of both robot arms $\rho$, a language goal $\mathbf{l} \in \{\ell_{as}, \ell_{sa}\}$, and an arm ID $\xi \in \{0, 1\}$, and the task is to predict an action. During training, the language goal is given in the data, but during evaluation, we use VLMs to determine which language goal, $\ell_{as}$ or $\ell_{sa}$, to use based on the given task. If the language goal is $\ell_{as}$, we assign the left arm ($\xi = 0$) to the acting policy and the right arm ($\xi = 1$) to the stabilizing policy, and conversely for $\ell_{sa}$. This allows our method to learn to map the appropriate acting or stabilizing actions to a given arm during training. Note that the predicted arm ID is discarded during evaluation. PerAct uses value maps to represent different components of the action space, where predictions for each arm are $\mathcal{Q}$-functions with state-action values. Formally, we have the following five value maps *per arm*, as the output of the arm's learned deep neural network, where:

$$
\begin{aligned}
&\mathcal{V}_{\text{trans}}^{\text{arm}} = \text{softmax}(\mathcal{Q}_{\text{trans}}^{\text{arm}}((x, y, z)|\mathbf{v}, \rho, \mathbf{l}, \xi)) \quad \mathcal{V}_{\text{rot}}^{\text{arm}} = \text{softmax}(\mathcal{Q}_{\text{rot}}^{\text{arm}}((\psi, \theta, \phi)|\mathbf{v}, \rho, \mathbf{l}, \xi)) \\
&\mathcal{V}_{\text{open}}^{\text{arm}} = \text{softmax}(\mathcal{Q}_{\text{open}}^{\text{arm}}(\omega|\mathbf{v}, \rho, \mathbf{l}, \xi)) \quad\quad\quad \mathcal{V}_{\text{collide}}^{\text{arm}} = \text{softmax}(\mathcal{Q}_{\text{collide}}^{\text{arm}}(\kappa|\mathbf{v}, \rho, \mathbf{l}, \xi)) \\
&\mathcal{V}_{\text{id}}^{\text{arm}} = \text{softmax}(\mathcal{Q}_{\text{id}}^{\text{arm}}(\upsilon|\mathbf{v}, \rho, \mathbf{l}, \xi))
\end{aligned} \quad ,
$$

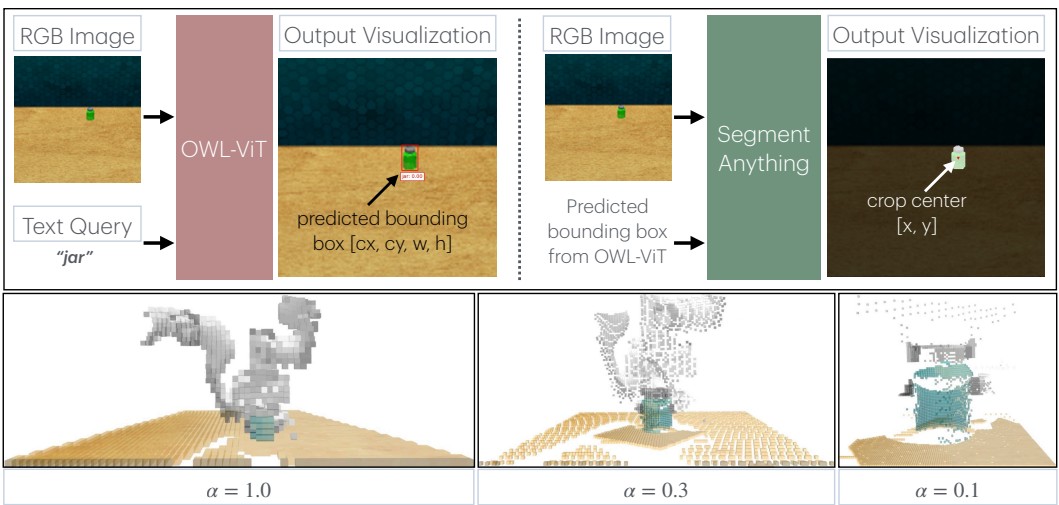

Figure 3: **Top**: VLMs usage as part of VoxAct-B, visualizing the `Open Jar` task in simulation, showing the role of OWL-ViT and Segment Anything. The RGB images from the front camera shown above are examples of actual (uncropped) images provided as input to the models. **Bottom**: visualization of different $\alpha$ values resulting in coarser grids ($\alpha = 1.0$) to finer grids ($\alpha = 0.1$). We use $\alpha = 0.3$ for `Open Jar`.

and where $(x, y, z)$, $(\psi, \theta, \phi)$, $\omega$, $\kappa$, and $\upsilon$ represent, respectively, the 3D position, the discretized Euler angle rotations, the binary gripper opening state, the binary collision variable, and the binary arm ID. At test time, to select each arm's action, we perform an "argmax" over all the input variables to the arm's five $\mathcal{Q}$-value, to get the five components. We refer the reader to [11] for more details.

The demonstrations provide labels for each arm's five action components, giving us the following nine label sources: $Y_{\text{trans}}^{\text{arm}} \in \mathbb{R}^{L \times W \times H}$ for translations, $Y_{\text{rot}}^{\text{arm}} \in \mathbb{R}^{(360/R) \times 3}$ (with $R = 5$ or 5-degree bins) for discretized rotations, $Y_{\text{open}}^{\text{arm}} \in \mathbb{R}^2$ for the binary open variables, $Y_{\text{collide}}^{\text{arm}} \in \mathbb{R}^2$ for the binary collide variables, and $Y_{\text{id}}^{\text{arm}} \in \mathbb{R}^2$ for the binary arm ID variables. The overall training loss for VoxAct-B is:

$$\mathcal{L}_{\text{total}} = \mathcal{L}_{\text{acting}} + \mathcal{L}_{\text{stabilizing}} \tag{1}$$

and where for both values of arm $\in \{\text{acting, stabilizing}\}$, we have

$$\mathcal{L}_{\text{arm}} = -\mathbb{E}_{Y_{\text{trans}}^{\text{arm}}}[\log \mathcal{V}_{\text{trans}}^{\text{arm}}] - \mathbb{E}_{Y_{\text{rot}}^{\text{arm}}}[\log \mathcal{V}_{\text{rot}}^{\text{arm}}] - \mathbb{E}_{Y_{\text{open}}^{\text{arm}}}[\log \mathcal{V}_{\text{open}}^{\text{arm}}] - \mathbb{E}_{Y_{\text{collide}}^{\text{arm}}}[\log \mathcal{V}_{\text{collide}}^{\text{arm}}] - \mathbb{E}_{Y_{\text{id}}^{\text{arm}}}[\log \mathcal{V}_{\text{id}}^{\text{arm}}],\tag{2}$$

which consists of a set of cross-entropy classifier-style losses for each component in the action.

### 4.2 VoxAct-B: Voxel Representations and PerAct for Bimanual Manipulation

When using voxel representations for fine-grained manipulation, a high voxel resolution is essential. While one can increase the number of voxels, this would consume more memory, slow down training, and adversely affect learning as the policy is optimizing over a larger state space. Therefore, given a voxel grid observational input $\mathbf{v}$ of size $(L \times W \times H)$ that spans $x^3$ meters of the workspace, we keep the number of voxels the same but reduce the relevant workspace. We use VLMs to detect the object of interest in the scene and "crop" the grid around this object, resulting in a voxel grid that spans $\alpha x^3$ meters of the workspace, where $\alpha$ is a fraction that determines the size of the crop. This allows zooming into the more important region of interest. The voxel resolution becomes $(\frac{L}{\alpha x}, \frac{W}{\alpha x}, \frac{H}{\alpha x})$ voxels/meters from the original resolution of $(\frac{L}{x}, \frac{W}{x}, \frac{H}{x})$ voxels/meters.

To detect the object of interest reliably, we use a two-stage approach similar to [16]. We input a text query and a RGB image from the front camera to OWL-ViT [64], an open-vocabulary object detector, to detect the object. Then, we use Segment Anything [65], a foundational image segmentation model, to obtain the segmentation mask of the object and use the mask's centroid along with point cloud data, obtained from the front camera's RGB-D image, to retrieve the object's pose with respect to the front camera. We use the pose of the object to determine the task-specific roles of each arm and the language goal. This cropped voxel grid and language goal are the input to our bimanual

manipulation policy. We call our method *VoxAct-B: Voxel-Based Acting and Stabilizing Policy*. See Figures 2 and 3 for an overview. Additional implementation details can be found in Appendix A.1.

## 5 Experiments

For simulation experiments, we build on top of RLBench [14], a popular robot manipulation benchmark widely used in prior work, including VoxPoser and PerAct. We extend it to support bimanual manipulation and introduce additional variations (see Appendix A.2 for details), making the tasks more challenging. We design the following four bimanual tasks:

- **Open Jar**: a jar with a screw-on lid is randomly spawned and scaled from 90% to 100% of the original size within the robot's $0.43 \times 0.48$ meters of workspace. The robot must grasp the jar with one hand and use the other to unscrew the lid in an anti-clockwise direction until it is removed.
- **Open Drawer**: a drawer is randomly spawned inside a workspace of $0.65 \times 0.91$ meters. It is randomly scaled from 90% to 100% of its original size, and its rotation is randomized between $-\frac{\pi}{8}$ and $\frac{\pi}{8}$ radians. The robot needs to stabilize the top of the drawer with one hand and then open the bottom drawer with the other.
- **Put Item in Drawer**: a drawer (the same type from Open Drawer) is randomly spawned in a workspace of $0.65 \times 0.91$ meters, and is randomly scaled and rotated using the same sampling ranges from Open Drawer. The robot needs to open the top drawer with one hand, grasp the item placed on top of the drawer with the other hand, and place it in the top drawer.
- **Hand Over Item**: a block is randomly spawned in a workspace of $0.43 \times 0.48$ meters. The robot needs to grasp a block with one hand and hand it over to the other.

See Figure 1 for an illustration. In the real world, we test Open Jar and Open Drawer using a coffee jar with dimensions $3.35 \times 2.85 \times 4.8$ inches and a drawer of dimensions $12 \times 12 \times 12$ inches. Note that the real-world jar and drawer cannot be opened without the use of a second arm.

### 5.1 Baselines and Ablations

In simulation, we compare against several strong baseline methods: **Action Chunking with Transformers (ACT)** [3], **Diffusion Policy** [15], and **VoxPoser** [16]. ACT is a state-of-the-art method for bimanual manipulation. Diffusion Policy represents the policy as a conditional denoising diffusion process and excels at learning multimodal distributions. ACT and Diffusion Policy use joint positions for their action space instead of predicting end-effector poses as our method. We adapt the Mobile ALOHA repository for ACT and a CNN-based Diffusion Policy, and we tune their parameters (e.g., chunk size and action horizon) to improve performance. For VoxPoser, we write and tune their LLM prompts to work on our bimanual manipulation tasks using the VoxPoser repository. Additionally, we include a **Bimanual PerActs** baseline, which trains separate PerAct policies for the left and right arms, to show how a straightforward bimanual adaptation of a single-arm, state-of-the-art voxel-based method performs. It uses the same number of voxels, $100^3$, as the original PerAct. See the Appendix for further details. We also test the following ablations of VoxAct-B:

- **VoxAct-B w/o VLMs**: does not use the VLMs to detect the object of interest and crop the voxel grid. It uses the same number of voxels as our method and the default workspace dimensions.
- **VoxAct-B w/o Segment Anything**: uses the bounding box obtained from OWL-ViT to compute the object's centroid.
- **VoxAct-B w/o acting and stabilizing formulation**: trains a left-armed policy for left arm actions and a right-armed policy for right arm actions. Otherwise, it is the same as VoxAct-B.
- **VoxAct-B w/o arm ID**: disables the arm ID loss function.

### 5.2 Experiment Protocol and Evaluation

To generate demonstrations in simulation, we follow the convention from RLBench and define a sequence of waypoints to complete the task, and use motion planning to control the robot arms

to reach waypoints. We generate 10 and 100 demonstrations of training data. Half of this data consists of left-acting and right-stabilizing demonstrations, and the other half contains right-acting and left-stabilizing demonstrations. We generate 25 episodes of validation and test data using different random seeds. We train and evaluate all methods using five random seeds and average the results. We evaluate all methods on the same set of test demonstrations for a fair comparison. See Appendix A.1 for details on how checkpoint selection is done in VoxAct-B.

In the real world, we use a dual-arm CB2 UR5 robot setup. Each arm has 6-DOFs and has a Robotiq 2F-85 parallel-jaw gripper. We collect ten demonstrations for each task with the GELLO teleoperation interface [66] for policy training. We use a flat workspace with dimension 0.97 m by 0.79 m and mount an Intel RealSense D415 RGBD camera at a height of 0.42 m at a pose which reduces occlusions of the object. For evaluation, we perform 10 rollouts per task. In `Open Drawer`, the arms have fixed roles of right acting and left stabilizing, and the acting arm opens the top drawer. The drawer has variations of 10 cm in translations and 20° of rotations. In `Open Jar`, we conducted two experiments: one has the roles of the arms reversed and fixed and the other has unfixed roles of each arm. The jar has variations of 12 cm in translations. See Appendix B for more details.

## 6 Results

### 6.1 Simulation Results

**Comparisons with baselines.** Table 1 reports the test success rates of baselines and VoxAct-B. When we train all methods using ten demonstrations, VoxAct-B outperforms all baselines by a large margin. In a low-data regime, the discretized action space with spatial equivariance properties (as used in VoxAct-B and Bimanual PerActs)

| Method | Open Jar | | Open Drawer | | Put Item in Drawer | | Hand Over Item | |
|---|---|---|---|---|---|---|---|---|
| | 10 | 100 | 10 | 100 | 10 | 100 | 10 | 100 |
| Diffusion Policy | 4.8 | 21.6 | 4.8 | 5.6 | 2.4 | 4.8 | 0.0 | 0.0 |
| ACT w/Transformers | 4.0 | 30.4 | 12.8 | 28.0 | 8.8 | 44.8 | 1.6 | 7.2 |
| VoxPoser | 8.0 | 8.0 | 32.0 | 32.0 | 4.0 | 4.0 | 0.0 | 0.0 |
| Bimanual PerActs | 8.0 | - | 36.8 | - | 5.6 | - | 0.0 | - |
| VoxAct-B (ours) | **40.0** | **59.2** | **73.6** | **72.8** | **39.2** | **49.6** | **19.2** | **14.4** |

Table 1: Performance of different methods on bimanual manipulation tasks in simulation, based on 10 or 100 (task-specific) training demonstrations. We use five training seeds for all methods, and evaluate on the same 25 episodes of unseen test data using the best checkpoints from validation (Section 5.2). The results are the average evaluation over five seeds. We only test Bimanual Peracts with ten demonstrations (not 100) due to computational constraints. VoxPoser does not have training, so its 10 and 100 results are identical.

may be more sample-efficient and easier for learning-based methods compared to methods that use joint space (ACT and Diffusion Policy). When we train all methods using more demonstrations (100), VoxAct-B still outperforms all baselines. Through ablations of ACT and Diffusion Policy, we found that removing environment variations greatly improved their performance. We attribute the tasks' difficulty to the following: high environment variation, difficult bimanual manipulation tasks with high-dimensional action spaces and fine-grained manipulation, and the two types of training data that each method needs to learn (based on which arms are acting and stabilizing).

Qualitatively, baseline methods, especially VoxPoser, typically struggle with precisely grasping objects such as drawer handles and jars. The baselines also struggle with correctly assigning the roles of each arm. For instance, a policy intended to execute acting actions may unpredictably produce stabilizing actions. Furthermore, they can generate kinematically infeasible actions or actions that lead to erratic movements, as seen in ACT and Diffusion

| Method | Open Drawer |
|---|---|
| VoxAct-B w/o VLMs | 19.2 |
| VoxAct-B w/o Segment Anything | 67.2 |
| VoxAct-B w/o acting and stabilizing | 64.8 |
| VoxAct-B w/o arm ID | 68.0 |
| VoxAct-B (ours) | **73.6** |

Table 2: Ablation experiment results in simulation.

Policy, which may be caused by insufficient training data. In contrast, we observe fewer of these errors with VoxAct-B.

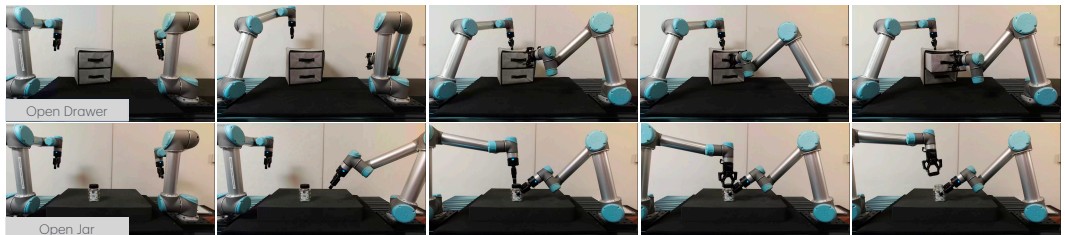

Figure 4: Example successful rollouts (one per row) of VoxAct-B on a real-world bimanual setup with UR5s.

**Ablation experiments.** Table 2 reports results on `Open Drawer` in simulation, based on 10 training demonstrations and evaluated across five training seeds. We use the same training and evaluation protocols as Table 1. VoxAct-B w/o VLMs performs poorly versus VoxAct-B because, without using the VLMs to reduce the physical space, the voxel resolution is lower due to the large workspace area for each individual voxel, which hinders fine-grained manipulation. VoxAct-B w/o Segment Anything performs worse than VoxAct-B because the predicted object centroid from Segment Anything is closer to the ground truth centroid obtained from the simulator, which is used for training VoxAct-B, than OWL-ViT's. We found that the closer the predicted object centroid is to the ground truth, the better the policy performs. Moreover, VoxAct-B w/o acting and stabilizing and VoxAct-B w/o arm ID perform worse than VoxAct-B, and they struggle with the same issues as the baselines. See Appendix A for additional experiments and details.

### 6.2 Physical Results

Figure 4 shows real-world examples of VoxAct-B. In `Open Drawer`, success is when the stabilizing arm holds the drawer from the top while the acting arm pulls the top part. VoxAct-B succeeds in 6 out of 10 trials; the failures include robot joints hitting their limits, imprecision in grasping the handle, and collisions with the drawer. In `Open Jar`, a success is when the stabilizing arm grasps the jar while the acting arm unscrews the lid. For the experiment with fixed roles of each arm, VoxAct-B succeeds in 5 out of 10 trials. While the stabilizing arm performs well in grasping the jar (9 out of 10 successes), the acting arm struggles with unscrewing the lid, succeeding only 5 out of 10 times due to imprecise grasping of the lid. For the experiment with unfixed roles of each arm, we train VoxAct-B on 10 left-acting, right-stabilizing and 10 right-acting, left-stabilizing demonstrations. It succeeds in 5 out of 10 trials, demonstrating its ability to learn from multi-modal, real-world data.

### 6.3 Limitations and Failure Cases

VoxAct-B implicitly assumes the object of interest does not encompass most of the workspace. If it does, it will be difficult to crop the voxel grid without losing relevant information. Another limitation is that VoxAct-B depends on the quality of VLMs. We have observed that some failures come from poor detection and segmentation from VLMs, which causes VoxAct-B to output undesirable actions. In addition to common errors described in Section 6.1, for `Put Item in Drawer`, VoxAct-B tends to struggle more with executing acting actions (e.g., drawer-opening and cube-picking/placing actions) in contrast to stabilizing actions.

## 7 Conclusion

In this paper, we present VoxAct-B, a voxel-based, language-conditioned method for bimanual manipulation. We use VLMs to focus on the most important regions in the scene and reconstruct a voxel grid around them. This approach enables the policy to process the same number of voxels within a reduced physical space, resulting in a higher voxel resolution necessary for accurate, fine-grained bimanual manipulation. VoxAct-B outperforms strong baselines, such as ACT, Diffusion Policy, and VoxPoser, by a large margin on difficult bimanual manipulation tasks. We also demonstrate VoxAct-B on real-world `Open Drawer` and `Open Jar` tasks using a dual-arm UR5 robot. We hope that this inspires future work in asymmetric bimanual manipulation tasks.

**Acknowledgments**

We thank our colleagues Gautam Salhotra, Kr Zentner, Yigit Korkmaz, Yunshuang Li, and Guangyao Shi for helpful writing feedback.

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

# A  Simulation Benchmark for Bimanual Manipulation

We chose RLBench [14] as our choice of simulator since it is well-maintained by the research community and has been used in a number of prior works [13, 12, 67, 68, 69, 16], including PerAct [11], which is a core component of VoxAct-B.

## A.1  Additional Implementation Details

VoxAct-B uses a voxel grid size of $50^3$ that spans $2^3$ meters. The proprioception data includes: the gripper opening state of both arms, the positions of the left arm left finger, left arm right finger, right arm left finger, right arm right finger, and timestep. Following PerAct [11], we apply data augmentations to the training data using SE(3) transformations: $[\pm 0.125\,\mathrm{m}, \pm 0.125\,\mathrm{m}, \pm 0.125\,\mathrm{m}]$ in translations and $\pm 45°$ in the yaw axis. We use 2048 latents of dimension 512 in the Perceiver Transformer [70] and optimize the entire network using the LAMB [71] optimizer. We use $\alpha = 0.3$ for Open Jar and $\alpha = 0.4$ for the drawer tasks and Hand Over Item. We select these $\alpha$ values by using a starting state of the environment with the largest scaling size factor for the object of interest and checking whether the object remains entirely contained in the voxel grid after cropping.

Deciding the best checkpoints for VoxAct-B and ablations is nontrivial since iterating over all possible combinations is computationally expensive. For example, with 400,000 training steps, using the same 10,000 checkpoint interval means there are $40 \times 40 = 1600$ possible combinations. Therefore, with the validation data, we use the latest stabilizing checkpoint to evaluate all acting checkpoints; we use the best acting checkpoint to evaluate all stabilizing checkpoints. Then, we use the best-performing acting and stabilizing checkpoints to obtain the test success rate.

The policy is trained with a batch size of 1 on an Nvidia 3080 GPU for two days. Note that the batch size is not optimized based on GPU memory capacity. The batch size can be increased to 6 using an Nvidia RTX Titan with 24 GB of VRAM. Moreover, VoxAct-B is much more efficient than Bimanual PerActs, requiring only two days of training to reach one million training iterations, while it would take Bimanual PerActs ten days to reach the same number of training iterations.

During policy evaluation, the language goal, $\ell_{as}$ or $\ell_{sa}$, is determined by VLMs based on the given task. Specifically, in Open Drawer and Put Item in Drawer, we use the drawer's pose relative to the front camera to determine which robot arm the drawer is facing. If it faces the left robot arm, $\ell_{as}$ is selected because the orientation gives the left (acting) arm a better angle for opening the drawer, and vice versa for the right robot arm. In Open Jar, we use the jar's pose to determine which robot arm it is closer to. If it is closer to the left robot arm, $\ell_{as}$ is selected because it provides the right arm with a better angle for grasping the jar based on our experience with this task. This also applies to the block in Hand Over Item.

## A.2  Additional Simulation and Task Details

We extend RLBench to support bimanual manipulation by incorporating an additional Franka Panda arm into the RLBench's training and evaluation pipelines. Importantly, we do not modify the underlying APIs of CoppeliaSim (the backend of RLBench) to control the additional arm; consequently, the robot arms cannot operate simultaneously, resulting in a delay in their control. However, this limitation is acceptable as our tasks do not require real-time, dual-arm collaboration.

Moreover, we modify Open Jar, Open Drawer, and Put Item in Drawer to support bimanual manipulation: (1) adding an additional Franka Panda arm with a wrist camera; (2) adding new waypoints for the additional arm; (3) adjusting the front camera's position to capture the entire workspace; (4) removing the left shoulder, right shoulder, and overhead cameras. The new tasks use a three-camera setup: front, left wrist, and right wrist. We also modify the data generation pipeline to use motion planning with the new waypoints, process RGB-D images and the new arm's proprioception data (joint position, joint velocities, gripper open state, gripper pose), and include the $[x, y, z]$ position (world coordinates) of the object of interest. These modifications also apply to the Hand Over Item task.

The success conditions of these tasks have also been modified: for `Open Jar`, we define a proximity sensor in the jar bottle to detect whether an arm has a firm grasp of the jar (the gripper's opening amount is between 0.5 and 0.93); for `Open Drawer`, we define a proximity sensor on the top of the drawer to detect whether an arm is stabilizing the drawer. While a robot arm could still "open" the drawer without the other arm's stabilization, we would not classify it a success in `Open Drawer`. Similarly, for `Open Jar`, a robot arm could "open" the jar lid without a firm grasp on the jar from the other arm, but this would not be classified as a success.

For the additional task variations, each method must learn from multi-modal demonstrations. For example, given ten training data, half have left-acting and right-stabilizing demonstrations, and the other half have right-acting and left-stabilizing demonstrations. During evaluation, the drawer from the drawer tasks is randomly rotated to face the left arm in the first half of the episodes and the right arm in the last half. This variation can make it kinematically infeasible for the left or right robot arm to open the drawer, requiring each method to determine the appropriate arm for acting and stabilizing. Additionally, we randomly scale objects (drawers and jars) from 90% to 100% of their original size.

### A.3 Multi-Task Experiment

We train a single policy of ACT and VoxAct-B on `Open Jar`, `Open Drawer`, and `Put Item in Drawer`, with 10 demonstrations for each task. For evaluation, both methods use the checkpoint with the best average validation success rate across all tasks. We use the same validation and test data as the baseline comparison. Table 3 presents the multi-task results, showing that VoxAct-B outperforms ACT on all tasks.

| | Open Jar | Open Drawer | Put Item in Drawer |
|---|---|---|---|
| ACT w/Transformers | 2.7 | 12.0 | 14.7 |
| VoxAct-B (ours) | **21.3** | **62.7** | **17.3** |

Table 3: Multi-task results of ACT and VoxAct-B trained on 10 demonstrations of each task and evaluated across three training seeds.

### A.4 Automatic Selection of $\alpha$

Instead of using a predefined $\alpha$, we use the VLMs to automatically determine the value of $\alpha$ by computing the largest dimension of the object of interest, with a small padding added. The method using the estimated $\alpha$ achieved a success rate of 32% on `Open Drawer`, compared to 73.6% when using the predefined $\alpha$, evaluated across three training seeds. We suspect the performance drop is due to the additional voxel resolutions the method must learn. As $\alpha$ varies between demonstrations due to the rescaling of objects, voxel resolution ($\frac{L}{\alpha x}, \frac{W}{\alpha x}, \frac{H}{\alpha x}$) also changes, making policy learning more challenging.

### A.5 Jars and Drawers of Different Appearances

We introduce two new jars and drawers of different appearances. In the new `Open Jar` experiment (i.e., different from the `Open Jar` experiment in the main paper), each method is trained on 10 demonstrations using 3 different jars. The two new jars are imported from the PartNet-Mobility Dataset [72]. Each method is then evaluated on 25 episodes of validation and test data using these jars across three training seeds. ACT obtains a success rate of 0%, while our method achieves a success rate of 41.3%. In the new `Open Drawer` experiment, each method is trained on 10 demonstrations using a drawer with 3 different textures, as we were unable to find a drawer with similar structures online. This experiment follows the same setup as the new `Open Jar` experiment. In summary, ACT achieves a success rate of 9.3%, while our method achieves a success rate of 33.3%.

### A.6 Ablation: Single Camera Setup

Instead of using a three-camera setup, we use a (front) single-camera setup on `Open Drawer`. In this setup, as the end-effector approaches the drawer handle, the view may be obstructed by the robot

arm. However, with the addition of a wrist camera, the handle remains visible, allowing the voxel grid to be constructed with a clear view of the drawer handle. This ablation tests how well VoxAct-B handles occlusion with a limited number of camera views. With a single-camera setup, VoxAct-B achieves a success rate of 60%, while using the original camera setup (front, left wrist, and right wrist) yields a success rate of 73.3%, evaluated across three training seeds.

## B  Real-World Experimental Details

**Hardware Setup.** An overview of the hardware setup is described in Section 5.2. Our perception system utilizes the D415 camera to capture RGB and depth images at a resolution of $1280 \times 720$ pixels, where the depth images contain values in meters. We apply zero-padding to these images, resulting in a resolution of $1280 \times 1280$ pixels. Hand-eye calibration is performed to determine the transformation matrices between the camera frame and the left robot base frame, as well as between the camera frame and the right robot base frame, using the MoveIt Calibration package. We use the python-urx library to control the robot arms. Additionally, I/O programming is employed to control the Robotiq grippers, as CB2 UR5 robots do not support URCaps.

**Data Collection.** We utilize the GELLO teleoperation framework to collect real-world demonstrations. Due to the lack of Real-Time Data Exchange (RTDE) protocol support in CB2 UR5s, a noticeable lag is present when operating the GELLO arms. For Open Jar, a dedicated function controls the gripper's counterclockwise rotations for unscrewing the lid and lifting it into the air, mitigating the instability caused by latency. This function is triggered when the operator activates the GELLO arm's trigger. Additionally, we found that fixing the stabilizing arm while the acting arm is in motion is crucial for effective policy learning, as it eliminates noise introduced by unintentional, slight movements of the stabilizing arm. Observations are recorded at a frequency of 2 Hz.

**Training and Execution.** For training, we use a higher value for stopped_buffer_timesteps, a hyper-parameter that determines how frequently keyframes are extracted from the continuous actions based on how long the joint velocities have been near 0 and the gripper state has not been changed, in PerAct's keyframe extraction function to account for the slower movements of the robot arms due to latency compared to simulation. We apply the inverse of the transformation matrices obtained from hand-eye calibration to project each arm's gripper position to the camera frame. Using the camera's intrinsics and an identity extrinsic matrix, we construct the point cloud in the camera frame, allowing both arms' gripper positions and the voxel grid to reside in the same reference frame. For evaluation, we multiply the transformation matrices from hand-eye calibration by the policy's predicted left and right gripper positions to obtain the tool center point positions in their respective robot base frames. We visualize each robot arm's predicted gripper position in Open3D before executing it on the robot. Additionally, we conduct real-world experiments using a Ubuntu laptop without a GPU, resulting in significantly slower policy inference and robot execution times—from capturing an image observation to moving the robot arms—compared to a GPU setup. Consequently, this results in longer pauses between each robot execution and extended real-world videos, as demonstrated on our website.

## C  Additional Implementation Details for the Baselines

We carefully tune the baselines and include the hyperparameters used in Table 4. *We report the results of the best-tuned baselines in Table 1.* For our three tasks, we found that for ACT, a chunk size of 100 worked well, consistent with the findings reported in [3]. The temporal aggregation technique did not improve performance in our tasks, so we disabled this feature. For Diffusion Policy, lower values (e.g., 16) of the action prediction horizon were inadequate, leading to agents getting stuck at certain poses and failing to complete the tasks, so we used an action prediction horizon of 100. We found the Time-series Diffusion Transformer to outperform the CNN-based Diffusion Policy on Open Drawer and Open Jar, while both of them achieved comparable success rates on Put Item in Drawer. We use a batch size of 32 for both methods, and the observation resolution is $128 \times 128$

| Hyperparameter | ACT Value | Diffusion Policy Value |
|---|---|---|
| learning rate | 3e-5 | 1e-4 |
| weight decay (for transformer only) | - | 1e-3 |
| # encoder layers | 4 | - |
| # decoder layers | 7 | - |
| # layers | - | 8 |
| feedforward dimension | 3200 | - |
| hidden dimension | 512 | - |
| embedding dimension | - | 256 |
| # heads | 8 | 4 |
| chunk size | 100 | 100 |
| beta | 10 | - |
| dropout | 0.1 | - |
| attention dropout probability | - | 0.3 |
| train diffusion steps | - | 100 |
| test diffusion steps | - | 100 |
| ema power | - | 0.75 |

Table 4: Combined hyperparameters of ACT and Diffusion Policy. A dash ("-") indicates the absence of a hyperparameter for a given method.

| Method | Open Jar | | Open Drawer | | Put Item in Drawer | |
|---|---|---|---|---|---|---|
| | FAS | FAS+NSV | FAS | FAS+NSV | FAS | FAS+NSV |
| Diffusion Policy | 20.8 | 40.8 | 24.0 | 46.4 | 14.4 | 19.2 |
| ACT w/Transformers | 31.2 | 56.0 | 28.8 | 35.2 | 34.4 | 75.2 |

Table 5: Ablation results of ACT and Diffusion Policy trained on 100 demonstrations and evaluated across five training seeds. "FAS" refers to the demonstrations with fixed acting and stabilizing arms (i.e., right acting and left stabilizing), while "FAS+NSV" refers to fixed acting and stabilizing and without size variation in the environment. We use the same validation and test data as the baseline comparison.

(same as VoxAct-B). For Diffusion Policy, we use the same image augmentation techniques as in [3]. As shown in Table 5, the performance of ACT and Diffusion Policy progressively improves as more environment variations are removed. For VoxPoser, we modified the LLM prompts to work with our bimanual manipulation tasks. See our VoxPoser prompts for details. For Bimanual PerActs, we deliberately chose to use $100^3$ voxels instead of the $50^3$ voxels used in VoxAct-B. The increased number of voxels provides higher voxel resolution, which is essential for fine-grained bimanual manipulation. This is demonstrated in the VoxAct-B w/o VLM ablation, which only utilizes $50^3$ voxels and shows a huge drop in performance compared to VoxAct-B.

