# OpenReview forum: "VoxAct-B: Voxel-Based Acting and Stabilizing Policy for Bimanual Manipulation"
_robot-learning.org/CoRL/2024/Conference — CoRL 2024_

### Official Review · Reviewer_6SQW · 2024-07-17
**Good paper, willing to increase score if one concern is addressed.**

**Originality:** 2
**Technical Quality:** 3
**Clarity Of Presentation:** 4
**Potential Impact:** 2
**Recommendation:** 2
**Confidence:** 5

**Review:**

1. Introduction is misleading. It is mentioned a couple of times how baselines, especially primitive based, demonstrate limited generalization and the reader is left to believe the paper attempts to address this. Yet the paper is evaluated on a very limited set of tasks, also primitives. Suggest some rework of introduction.

2. The two last contribution bullets are not really contributions. A paper should always be accompanied by supporting experiments. Suggest removing those.

3. Ln. 81-82: “...have reduced the computational cost of PerAct by avoiding voxel representations but often need multiple views of the scene to achieve optimal performance”, re: this method must also need multiple views in order to construct a voxel grid. The voxel grid is assumed to be part of the observation but to have that, multiple views would still be required.

4. Ln. 129-130: “Hence, either arm can execute an acting policy or a stabilizing policy, which improves policy learning”, re: not intuitive why this would improve policy learning. Sounds like the learning problem becomes more multi modal, if anything, which intuitively would make it harder.

5. VLM specifics (which model, input form, output form, constraints to be able to output something interpretable to l_as or l_sa, how that gets interpreted to l_as or l_sa) should have been discussed earlier. Reader is left with many assumptions in the method section.

6. Rotation discretization over 5 bins means predicting bins of 72 degrees each. Bins this coarse can’t give good results on precision tasks where rotation matters.

7. Paper assumes language instructions that only contain one single interaction object. Method does not generalize to free form language instructions. Assumptions should be stated more explicitly.

8. alpha hyperparameter for grid size resolution varies per task and is selected via a trial and error heuristic. Not great for generalizability.

9. Method still requires training one policy per task for 2 days and batch size of 1 implies large computation requirements.

10. Ln. 264-265: Fixing the roles of stabilizing and acting arm in ACT and Diffusion policies improving their performance, re: this is a very interesting observation that would be useful to the community. Suggest adding these numbers in the appendix for the readers.

11. Not convinced about how much voxelizing adds vs having the same method predicting 3D points instead. This is a strong ablation to add.

12. Nit: figure 4 hyperlink doesn’t work.

13. Asymmetric bimanual manipulation is used as a term. Suggest defining it a bit more formally to help the reader.

14. Generally, experiments are weak. Method is trained on 3 SIM tasks (plus 2 real tasks) and evaluated on the same tasks while a different policy needs to be trained on each task separately. Therefore, very limited (if any) generalization is shown.

**Quality Of The Limitations Section:**

3

**Questions For Rebuttal:**

1. No skills library? Method does mention the use of primitives. That is in a sense a skills library?

2. Isn’t always assuming a stabilizing and an acting arm already hurting generalization?

3. Ln. 115-116: how do you keep important keyframes of a demonstration such as interaction points? In those time steps, the gripper state does not remain unchanged often.

4. What VLM are you using? How do you constrain it to output language in the form of e.g. “Hold the desk with the left arm and open the bottom drawer with the right” which is my assumption, otherwise it’s not clear how you map language to l_as or l_sa.

5. Ln. 139-140: “Note that the predicted arm ID is discarded”, re: I’m assuming the 2 predicted next voxels for each arm (stabilizing and acting one) would differ depending on whether the left arm is the stabilizing and the right the acting or vice versa. If you are discarding the predicted arm ID, how do you discern between the two possible predicted modes at inference?

6. Why R = 5 for rotation bins? Discretization error must be too large to perform precision tasks.

7. Fig. 3: Why do you need Segment Anything at all here? If Owl-ViT gives reliable enough bounding boxes, then your crop center would be the center of your bounding box and that should be close enough?

8. Ln. 165-166: Point cloud data access/usage has not been introduced so far. You get this from one RGB-D camera? How do you get complete enough point clouds to get a 3D object centroid (which I assume you need)?

9. Ln. 234-236: Unless I’m misunderstanding, with this way of picking checkpoints, you may still be missing the best combination of stabilizing-acting checkpoints you oculd possibly have?

10. In the real world setup, you train each task policy on 10 demonstrations for that task, correct?

11. Nit: Does your Open Drawer setting actually require stabilization at all to succeed? I can’t really tell what the friction is like. Does the drawer furniture actually move if you don’t stabilize it at all and only pull the drawer?

**Robotics Focus:**

4

**Summary Of Paper:**

The paper proposes the use of VLMs as a means to reduce the state space to voxelize in order to learn bimanual policies.

**Summary Of Recommendation:**

Thank you to the authors for this work. The paper was well written and clear overall. Main reason for the recommendation is that the experiments show limited generalization. Each task needs to be trained as a separate policy and evaluation is done essentially in distribution. However, there is value for the community to read this paper. I’m very willing to increase my score if authors include an experiment that showcases some generalization e.g. training one multi-task policy for all tasks and showcasing that the policy can perform well not only on in distribution jars/drawers but also jars/drawyers of different appearance more than just color and minimal rescaling.

---

### Official Review · Reviewer_7y5C · 2024-07-21
**Review of VoxAct-B**

**Originality:** 4
**Technical Quality:** 5
**Clarity Of Presentation:** 4
**Potential Impact:** 4
**Recommendation:** 4
**Confidence:** 4

**Review:**

Bi-manual manipulation tasks is a very novel field and this paper investigates it further by leveraging VLMs and reconstructing a voxel map of the environment. We have seen bi-manual manipulation in previous work but utilizing robots like URs are novel to the field and interesting to see commodity hardware being used for such purposes. Involving a VLM for planning and classifying the actor and stabilizer. The methodology and the evaluations between the paper and the baselines were well detailed and thorough. Although, I would like to see some generalization and scalability in place since simulation and sim2real is happening, it would show a better use of the paper and the goal it is trying to achieve. I think this might be achievable by introducing multiple objects and different variations of the same object in the scene. It might also be fruitful to show this policy being run in different backgrounds and with different lightings to show that the real world variations are not significantly changing the results.

**Quality Of The Limitations Section:**

2

**Questions For Rebuttal:**

As mentioned above, it might be useful for more experiments in different environments and with various objects.

**Robotics Focus:**

4

**Summary Of Paper:**

This paper introduces a language-conditioned, voxel-based method for efficient and generalizable bimanual manipulation using Vision Language Models (VLMs)

**Summary Of Recommendation:**

I like the way this paper has explained how VLMs can be leveraged for task planning and allow scalability.

---

### Official Review · Reviewer_nMKS · 2024-07-26
**VoxAct-B Paper - Official Review**

**Originality:** 3
**Technical Quality:** 3
**Clarity Of Presentation:** 4
**Potential Impact:** 3
**Recommendation:** 3
**Confidence:** 5

**Review:**

### Strengths

- The paper extends RLBench to support bimanual manipulation for Open Jar, Open Drawer and Put Item in Drawer tasks. This is very interesting to foster language-guided bimanual manipulation tasks.
- The authors demonstrate that VoxAct-B can be used on real-robot experiments for open drawer and open jar tasks and provide videos of its execution.
- Leveraging VLMs to select the arms role and provide a voxel grid with more resolution around the object of interest seems to be a novel and nice idea.

### Weaknesses

- While the benchmark is interesting, I think that the number of proposed tasks falls a bit short (only three and Put Item In Drawer and Open Drawer are very similar). I wonder where this limitation comes from. Were there no more tasks inside the simulator that could benefit from bimanual manipulation?
- Related to the previous question. Is bimanual manipulation beneficial to solving the proposed tasks? Do we need bimanual manipulation? Does not bimanual manipulation overcomplicate the task's difficulty? I can see that other works doing one arm manipulation have better performance on the given tasks: Act3D [1] Open Drawer SR=92 (10 demos) / 93 (100 demos) and Put Item in Drawer SR=82.0 (10) / 90.0 (100), RVT-2 [2] Open Drawer SR=74.0 (100 demos) and Put Item in Drawer SR=96.0 (100 demos), RVT [3] Open Drawer SR=71.2 (100 demos) and Put Item in Drawer SR=88.0 (100 demos), etc. I think finding tasks more suitable for bimanual manipulation is necessary to make the paper's motivation stronger. Maybe, the motivation is that opening a jar or drawer might not be possible without a second arm.
- Is this approach suitable for multi-task settings (one model trained on all the tasks)? I am not sure whether the results presented in Table 1 are for single-task or multi-task settings. This should be clarified.
- I wonder why the authors don’t test on Put Item in Drawer task on the real robot. Is there any limitation?
- While using a VLM is a nice idea. I wonder how this compares with an approach similar to Act3D [1] where action is predicted in a coarse-to-fine manner. Could we remove the VLM and predict an action with the complete voxel grid, and then use this action to create a higher resolution voxel grid around the predicted action?
- Using the centroid of the object mask + a predefined $\alpha$ where you need to check whether the object is completely contained in the cropped voxel grid doesn’t see the most optimal way to approach it. Wouldn’t it be possible to compute the pose for the max and min object mask and use it to crop the voxel grid in a more task-generalized way?

[1] Act3D: 3D Feature Field Transformers for Multi-Task Robotic Manipulation. CoRL 2023, Gervet et al.

[2] RVT-2: Learning Precise Manipulation from Few Examples. RSS 2024, Goyal et al.

[3] RVT: Robotic View Transformer for 3D Object Manipulation. CoRL 2023, Goyal et al.

**Quality Of The Limitations Section:**

3

**Questions For Rebuttal:**

Please, take a look at the weaknesses in the review. Questions should be addressed to make the paper stronger.

**Robotics Focus:**

4

**Summary Of Paper:**

This paper proposes VoxAct-B, a language-guided, voxel-based method that uses VLMs to perform precise bimanual robotic manipulation. Given RGB images of the scene and a text query, VoxAct-B uses OWL-ViT as an open-vocabulary object detection algorithm to predict a bounding box around the object of interest. Then, VoxAct-B uses SAM to predict a segmentation mask for the object given its predicted bounding box and uses the mask centroid along with the point cloud data to compute the object’s pose to the camera. This pose serves multiple purposes: 1) Determine the role of each arm (either acting on the object or stabilizing) and thus the correct language instruction, and 2) Crop the whole scene voxel grid into a fine-grained voxel grid around the object of interest to have better resolution, thus better precision, at a lower computational cost. The language instruction and fine-grained voxel grid are given to a learned bimanual manipulation policy to predict each arm's next pose, gripper openness, and whether the motion planner should check for collision. Finally, the approach is evaluated in an extended version of RLBench for bimanual manipulation and the real robot.

**Summary Of Recommendation:**

My recommendation is Weak Reject. I believe that studying bimanual manipulation is very interesting and that the robotics community could greatly benefit from an extension of the RLBench benchmark for bimanual manipulation. VoxAct-B is novel enough and the authors demonstrate it works well on the real robot. However, I have some concerns about the benchmark quality and if the paper's motivation is strong enough, and the pointed weaknesses should be addressed.

---

### Author Rebuttal · Authors · 2024-08-13

Revised Manuscript

---

### Decision · Program_Chairs · 2024-09-04

**Decision:**

Accept

**Comment:**

Strengths:
+ The paper introduces simulated bimanual tasks for reproducible benchmarking.
+ VoxAct-B is validated with real-robot experiments.
+ The approach uses VLMs in a new way to isolate regions of interest for 3D manipulation.


Weaknesses:
- The number of bimanual tasks proposed in the benchmark is limited.
- Experiments do not highlight generalization capabilities of the approach. This includes multi-task and robustness (lighting and different backgrounds) tests.
- It’s unclear if some of the tasks (e.g. Put Item in Drawer) need the bimanual setting to solve the task at all.
- Some implementation details are missing, for e.g., how the VLM is used, and hyperparameters such as rotation discretization.

Post rebuttal:
After an extensive rebuttal by the authors, some concerns regarding the quantity and quality of tasks remain. However, 2/3 reviewers feel that the work is beneficial to the community.